# Appearance and Performance-Enhancing Drugs and Supplements, Eating Disorders Symptoms, Drive for Muscularity, and Sexual Orientation in a Sample of Young Men

**DOI:** 10.3390/nu14224920

**Published:** 2022-11-21

**Authors:** Ata Ghaderi, Elisabeth Welch

**Affiliations:** 1Division of Psychology, Department of Clinical Neuroscience, Karolinska Institutet, 17177 Stockholm, Sweden; 2Stockholm Health Care Services, Region Stockholm, Stockholm Centre for Eating Disorders, 10462 Stockholm, Sweden; 3Centre for Psychiatry Research, Department of Clinical Neuroscience, Karolinska Institutet, 17177 Stockholm, Sweden

**Keywords:** performance-enhancing drugs, performance-enhancing supplements, appearance-enhancing drugs and supplements, eating disorders, drive for muscularity, sexual orientation

## Abstract

In an anonymous online study (*N* = 824), we investigated the frequency of use of appearance and performance-enhancing drugs and supplements (APEDS) in a sample of young men (15–30 years) in Sweden, along with their self-reported eating disorder (ED) symptoms, drive for muscularity and sexual orientation. A total of 129 participants (16.1%) reported regular use of supplements (at least once a week), including one individual using anabolic steroids (0.1%), while a lifetime use of APEDS was reported by 32.3%. The overlap between those using protein supplements and creatine was large (83.6%). Some symptoms of ED (e.g., dietary restraint, objective binge eating, self-induced vomiting, and excessive exercise) significantly predicted the use of APEDS. In addition, the use of APEDS was significantly predicted by the drive for muscularity. The prediction was stronger for the behavioral component of drive for muscularity (*Exponential B* = 8.50, *B* = 2.14, *SE* = 0.16, *p* < 0.001, Negelkerke *R*^2^ = 0.517) than for its attitudinal component (*Exponential B* = 1.52, *B* = 0.42, *SE* = 0.06, *p* < 0.001, Negelkerke *R*^2^ = 0.088). A significantly larger proportion of those identifying as heterosexual reported using APEDS (34.4%) compared to those identifying themselves as homosexual (25.0%), bisexual (19.2%) or other (23.7%). Overall, our results suggest that the use of APEDS might be more related to the drive for muscularity and sexual orientation than symptoms of ED.

## 1. Introduction

Eating disorders (EDs) tend to have a chronic path and are associated with marked functional impairment, significant morbidity and increased mortality [1,2]. Although EDs are more prevalent among female, these conditions are a significant cause of morbidity and impairment among males as well [3]. Gorrell and Murray (2019) suggest that the prevalence of EDs for males, cited in recent history of research, is a gross underestimate [4]. As argued in a review of EDs among males [5], although the rate ratio of lifetime prevalence of main ED diagnoses among females versus men is often estimated to be 10:1, such figures probably reflect clinical under-detection of males. It is well known that male are less prone to seek professional help for EDs [6] due to individual, socio-environmental, organizational and contextual factors. Epidemiological studies generally show a more nuanced picture with clearly less extreme rate ratio of EDs among males versus female [7,8,9,10]. Given the historical focus of research on females, the nosology of ED is mainly based on female symptom profiles [11]. However, more recent studies suggest some similarities as well as differences between male and female ED etiology, risk factors, course and outcome of treatment [4]. As an example, there are substantial differences between ideal male and female body image. Although internalization of thin ideal has repeatedly been shown to be a significant risk factor for the emergence of ED in females [12,13], the body type that is often idealized and internalized among males centers on muscularity [4]. Such an ideal might be the driving force behind the drive for muscularity, and use of appearance and performance-enhancing drugs and supplements (APEDS) among males. Use of APEDS is growing into a public health concern [14]. Lifetime use of APEDS has also been related to a positive ED screen among college and university students regardless of sex [15]. Use of APEDS has also been related to EDs and muscle dysmorphia symptoms in cis-gender sexual minority men and women [16]. As EDs are more common among homosexual than heterosexual males [4,17], use of APEDS might also be different among males with minority sexual orientations compared to heterosexual males. Previous research suggests that sexual minority adolescents are at higher risk for the usage of anabolic steroids and attempts to gain weight [18]. In line with the minority stress theory [19], victimization experiences have also been linked to the use of anabolic steroid in sexual minority adolescent boys [20]. Likewise, heterosexist discrimination and sexual orientation concealment [19] has been associated with use of APEDS among sexual minority adults [21]. 

As the use of anabolic steroids are connected to some level of stigma, it is plausible to assume that an anonymous investigation might provide a less biased picture of their consumption. This might also be true for other sensitive issues such as sexual orientation and EDs among males. As an example, men with EDs perceive it as a “girl’s disease”—label attached to EDs [22]. The current study aims to contribute to the literature on EDs among men by investigating the relationship between use of APEDS and symptoms of EDs, drive for muscularity and sexual orientation in a sample of males, who were recruited through social media and participated in an anonymous Internet-based survey. As the inter-quartile range for the peak age of onset for EDs is 15–23 years [23], and to consider the time required to recover from EDs [24] and long latency in seeking professional help for EDs [25], we chose to recruit a sample of young males (15–30 years) in the current study. 

We hypothesized that the use of APEDS would have a positive correlation with symptoms of EDs and the drive for muscularity. We also hypothesized that men who identify as a member of a sexual minority group would report significantly more use of APEDS compared to heterosexual men. 

## 2. Methods

### 2.1. Participants

Detailed information on participants has been provided in a previously published work by Ghaderi et al. [17]. A total of 824 males out of 1817, who opted to commence the survey, were included in the analyses. Of the initial 1817, 37.8% (*n* = 687) did not complete any of the questionnaires after reading the initial information about the study. In addition, 306 were excluded due to not meeting the inclusion criteria (e.g., other sex than males, outside of the age range of the study). The mean age of the participants was 24.79 years (*SD* = 3.64, range: 15–30) and their mean body mass index (BMI) was 24.55 (*SD* = 4.74). Slightly more than half of the participants (55.3%) reported living in a large city in Sweden (defined as having more than 200,000 inhabitants). More than a third (37.4%) lived alone, 26.6% with a partner and 16.4% with parents. The rest reported living with a friend (14.4%) or other living conditions (5%). In terms of education, 64.3% reported having a university degree or studying at university, and the rest reported having completed high school (28%), or still being in junior high or high school (7.7%). In terms of occupation, the majority reported studying (49.6%), while 38% reported being employed and the rest reported being on sick leave (1.9%), parental leave (2.8%) or having other occupations (2.8%). As reported earlier [17], the high educational level of the sample and slightly lower rate of unemployment compared to the national level are in line with national statistics for people aged 16 and older who live in large cities in Sweden.

### 2.2. Procedure

To obtain data from Swedish speaking males (15 to 30 years), we used an anonymous online survey and invited the target group through posts on social media (mainly Facebook). By clicking a link included in the post, participants were directed to a secure web page presenting information about the study that included the aim of the study, its voluntary and anonymous nature including the deletion of IP addresses, the option to quit participation at any time, responsible handling of data and presentation of results on group level and ways to contact the researchers for more information. Those interested could then access the survey after reading the information. No ethical approval was obtained given the anonymous nature of the study and guidelines in Ethical Review Act concerning the Ethical Review of Research Involving Humans (2003:460), or the Personal Data Act (1998:204) in Sweden. 

### 2.3. Instruments

Demographic information was obtained by asking about sex, age, education, living condition and occupation, described in more detail elsewhere [17]. Use of APEDS were determined by asking the participants to report the frequency with which they used anabolic steroids, protein supplements or creatine on a scale. This includes the following six levels: (1) at least once a week, (2) at least twice a month, (3) at least once a month, (4) a few times per year, (5) less than few times/year, or (6) never.

Symptoms of EDs were measured by the sixth edition of the Eating Disorders Examination Questionnaire (EDE-Q) [26]. The EDE-Q has 28 items, and it provides both behavioral and attitudinal measures of ED, as well as a global score. For this report, we used the global EDE-Q score and its subscales (dietary restraint, eating concern, weight concern, and shape concern), as well as behavioral symptoms of EDs (e.g., regular occurrence of binge eating and self-induced vomiting). In addition, the EDE-Q was also coded according to Friborg et al. (2013) [27] with an alternative four-factor structure (dietary restraint, preoccupation and restriction, weight and shape concerns, and eating shame). This version has shown good psychometric properties in sexual minority groups [28]. The EDE-Q has acceptable to good psychometric properties, as reported previously [17]. Higher scores indicate more severe ED psychopathology. 

Drive for muscularity was measured by the Drive for Muscularity Scale [29], which is a 15-item measure that provides an attitudinal, a behavioral and an overall score. It possesses good psychometric properties [29], and its internal consistency in the current study was high (0.90). Higher scores indicate stronger drive for muscularity.

Information on sexual orientation was obtained by asking the participants to report their sexual orientation using one of the following categories: heterosexual, homosexual, bisexual or other. 

### 2.4. Statistical Analysis

Data were analyzed using descriptive statistics, Chi-square, and logistic regression in SPSS statistics (version 28).

## 3. Results

Two hundred and fifty-five participants (31.9%) reported lifetime use of protein supplement or creatine. In addition, three individuals reported using anabolic steroids (0.4%). The overlap between those using protein supplements and creatine was large (83.6%). Only five participants (0.6%) who used creatine did not report using protein, while 15.8% of those who used protein supplements did not use creatine. In the sample, 26.7% also reported using other natural supplements, mostly vitamins. The frequency of using supplements and steroids is presented in Table 1. 

The use of APEDS was significantly predicted by the dietary restraint subscale of the EDE-Q. The logistic regression model was significant *χ*^2^ (1, *n* = 799) = 24.17, *p* < 0.001, and it correctly classified 68.0% of the participants. However, it explained only 4.2% of the variance (Negelkerke *R*^2^). Those with higher restraint had more than 30% higher odds of using APEDS (*Odds ratio* = 1.33, 95%*CI* [1.19, 1.50], *B* = 0.14, *SE* = 0.07, *p* < 0.001). The use of APEDS was not significantly predicted by other subscales of the EDE-Q. The association between use of APEDS and the global EDE-Q was weak, and it failed to reach statistical significance (*Odds ratio* = 1.15, 95%*CI* [0.99, 1.33], *B* = 0.14, *SE* = 0.06, *p* = 0.055). When the EDE-Q was coded according to Friborg and colleagues [27], the use of supplements was still significantly predicted only by the dietary restraint subscale (*Odds ratio* = 1.23, 95%*CI* [1.13, 1.34], *B* = 0.21, *SE* = 0.04, *p* < 0.001), but not the other subscales (i.e., Preoccupation and Restriction, Weight and Shape concerns, and Eating Shame). The regression model for the dietary restraint subscale, based on Friborgs coding, was significant *χ*^2^ (1, *n* = 799) = 23.75, *p* < 0.001, and correctly classified 67.8% of the cases. However, the explained variance was low (Negelkerke *R*^2^ = 0.041). Rerunning the analyses based on sexual orientations showed that the prediction only emerged among those who identified as heterosexuals (*Odds ratio* = 1.24, 95%*CI* [1.13, 1.36], *B* = 0.21, *SE* = 0.05, *p* < 0.001). Overall, a significantly larger proportion (*χ*^2^ (3, *n* = 798) = 9.12, *p* = 0.028) of those identifying as heterosexual reported using APEDS (34.4%) compared to those identifying themselves as homosexual (25.0%), bisexual (19.2%) or other (23.7%). 

The use of APEDS was significantly predicted by objective binge eating (*Odds ratio* = 1.49, 95%*CI* [1.03, 1.16], *B* = 0.40, *SE* = 0.19, *p* = 0.035). The prediction model was significant *χ*^2^ (1, *n* = 791) = 4.35, *p* = 0.037. Although it classified 67.9% of the cases correctly, the explained variance in the model was less than one percent (Negelkerke *R*^2^ = 0.008). The use of APEDS was also predicted by self-induced vomiting (*Odds ratio* = 3.44, 95%*CI* [1.11, 10.62], *B* = 1.24, *SE* = 0.58, *p* = 0.032). The overall prediction model was significant *χ*^2^ (1, *n* = 791) = 4.76, *p* = 0.029), and 68.1% of the cases were classified correctly. However, the explained variance was low (Negelkerke *R*^2^ = 0.008) in this model as well. Finally, the use of APEDS was also predicted by excessive exercise (*Odds ratio* = 3.54, 95%*CI* [2.36, 5.37], *B* = 1.26, *SE* = 0.21, *p* < 0.001). This model was also significant *χ*^2^ (1, *n* = 791) = 35.42, *p* < 0.001, and it explained 6.1% of the variance (Negelkerke *R*^2^) and classified 70.1% of the cases correctly. 

The logistic regression model to predict the use of APEDS by the drive for muscularity was significant *χ*^2^ (1, *n* = 799) = 195.34, *p* < 0.001. It predicted 75.3% of the cases correctly and explained more than 30% of the variance (Negelkerke *R*^2^). The odds of using APEDS was more than three times higher among those with a higher drive for muscularity (*Odds ratio* = 3.37, 95%*CI* [2.76, 4.11], *B* = 1.22, *SE* = 0.10, *p* < 0.001). In the logistic regression models for behavioral subscale of the drive for muscularity, the odds for using ADEPS was more than 8 times larger (*Exponential B* = 8.50, 95%*CI* [6.23, 11.60], *B* = 2.14, *SE* = 0.16, *p* < 0.001). The model was significant *χ*^2^ (1, *n* = 799) = 368.86, *p* < 0.001, and accounted for more than half of all the variance (Negelkerke *R*^2^ = 0.517), while it classified 82.9% of the cases correctly. On the other hand, the prediction by the attitudinal component of the drive for muscularity showed a minor increase in odds (*Odds ratio* = 1.52, *B* = 0.42, 95%*CI* [1.35, 1.70], *SE* = 0.06, *p* < 0.001) in a significant model *χ*^2^ (1, *n* = 799) = 52.11, *p* < 0.001 that explained 8.8% of the variance and classified 68.6% of the cases. All the significant outcomes in the logistic regressions remained significant after controlling for age and BMI.

Finally, to control for the role of the drive for muscularity in the relationship between the use of APEDS and restraint, an additional logistic regression was run with drive for muscularity entered in the first block and restraint in the second. The model was significant *χ*^2^ (1, *n* = 799) = 195.38, *p* < 0.001. It explained 30.3% of the variance and classified 75% of the cases correctly. The drive for muscularity was a significant predictor (*Odds ratio* = 3.34, 95%*CI* [2.71, 4.11], *B* = 1.21, *SE* = 0.11, *p* < 0.001), but restraint did not significantly predict the use of supplements after controlling for the drive for muscularity (*Odds ratio* = 1.02, 95%*CI* [0.88, 1.17], *B* = 0.02, *SE* = 0.07, *p* = 0.84). 

## 4. Discussion

The use of APEDS was common among young males in the current study, and it was more related to the drive for muscularity and sexual orientation than symptoms of EDs. In a large national sample (*N* = 7401) of college and university students, including both sexes in the US (ages 18–30 years), the lifetime prevalence of the use of protein supplements and creatine was 23.8% and 7.7%, respectively [30]. The corresponding lifetime figures (Table 1) in our study of young males were 31.9% and 16.5%. Male sex was related to a greater likelihood of lifetime protein and creatin supplement use in the US study [30], which might partly explain larger prevalence figures in our study of young males. Focusing on male college students (18–26 years) from the top-10 National Collegiate Athletic Association Division I universities in the US, the rate of current use of appearance- or performance-enhancing supplements was 38.9% [31]. This is a highly selected sample in which athletic performance is assumably a top priority, and a very high current rate is expected. If “current rate” in our study is defined in terms of at least monthly use of these supplements, then it will amount to 23.1%. The anonymous nature of the study might increase the truthful report of use of such substances. It has been shown that computerized surveys leads to significantly more reporting of sensitive issues (e.g., socially undesirable behaviors) [32]. In a study on the role of anonymity and privacy in survey methodology, participants reported significantly higher mean comfort levels with anonymous surveys compared to non-anonymous [33]. Given the difficulties inherent in unknown representativity of Internet-based surveys, further replications using the same methodology are warranted to arrive at a robust estimate in use of appearance- and performance enhancing supplements among young males. 

Previous studies suggest an association between the use of APEDS and positive ED screen among college students [15], including cis-gender sexual minority groups [16]. Our study confirms these outcomes, but these relationships were not as consistent in our study as in previous investigations; the use of APEDS was only predicted by the dietary restraint subscale of the EDE-Q and some behavioral symptoms of ED (binge eating, vomiting and excessive exercise). It is important to note though, that despite significant predictions, the amount of explained variance was low. We also found a stronger relationship between the use of APEDS and the drive for muscularity. The use of APEDS could no longer be predicted by ED symptoms when controlling for the drive for muscularity, thus it may be plausible to assume that the use of APEDS may be more related to the drive for muscularity than ED symptoms. Alternatively, the EDE-Q items and its specific subscales, with the exception of dietary restraint, may be less efficient in capturing the nature of eating disorders in men, elicited by the drive for muscularity. Although the male body ideal, features a dual focus on drive for muscularity and leanness [34], muscular-oriented ED symptoms are considerably more common among men [35]. The strong relationship between the use of APEDS and the drive for muscularity among men is also in line with the suggested specific pattern of “bulking and cutting” within muscularity-oriented ED, as summarized by Gorrell and Murray [4], in reference to the observations made by Mosley [36]. This means alternating periods of consuming high amounts of protein, and periods of dietary restriction to lose body fat. While the second part (i.e., cutting: restraint) might be adequately captured by the EDE-Q, the bulking periods might not be as efficiently reflected in the EDE-Q items. Although the bulking may in some instances be similar to overeating or binge eating, it may be perceived, by the individual in search of increased muscularity, as intentional, controlled, and planned eating of a specific source of macronutrients. It remains a conceptual as well as empirical question whether bulking and cutting should be considered a facet of EDs in men, or dysfunctional behaviors elicited by the drive for muscularity. To address this question, future assessment of symptoms of EDs should include specific questions on the overuse of macronutrients and supplements, as well as restraint with focus on increased leanness. 

Cis-gender sexual minority boys and men seem to engage in muscularity-oriented behaviors, such as use of anabolic steroids [20] and protein-supplements [37]. The use of anabolic steroids was reported by only a few participants in our study, which does not allow further investigation of potential differences in sexual orientation among those using anabolic steroids. Interestingly, however, we found that those identifying as heterosexual reported significantly more use of supplements in general (protein and creatine) (34.4%) than those with homosexual (25.0%), bisexual (19.2%) or other sexual orientation (23.7%), which is in contrast to the observations and predictions of previous research [18,20,21]. Earlier research had more specific focus on the use of anabolic androgenic steroids among adolescents and men with different sexual orientation, and not overarching categories of APEDS. In further exploration of this question, attention should also be paid to the difference in motivation to use supplements. Murray and colleagues [38] showed that men who used anabolic steroids primarily due to appearance-related concerns reported greater overall ED psychopathology than those using anabolic steroids mainly for performance purposes. 

The findings of the current study should be interpreted with attention to its limitations. Although an anonymous Internet-based study may provide a better opportunity to obtain sincere answers to sensitive questions, its representativity remains unclear. Contrasting the demographics of the sample with nationwide data from Sweden shows that at best, our sample might be a fair representation of adolescents and young males in large cities in Sweden. In future replications, more attention should be paid to recruitment of participants from rural areas, and those with lower education to obtain as much representative samples as possible. As the eating psychopathology of males has some similarity and some differences to those of females, future investigations should consider adding specific questions (e.g., behaviors such as bulking and cutting) to the battery of instruments. Finally, the specific motivation for using APEDS should also be assessed. 

To conclude, 32.3% reported lifetime the use of APEDS. Some symptoms of EDs (e.g., restraint, objective binge eating, self-induced vomiting, and excessive exercise) significantly predicted the use of APEDS. The use of APEDS was significantly associated with drive for muscularity. Finally, in contrast to previous observations and predictions of previous research, a larger proportion of those identifying as heterosexual reported using APEDS compared to those with minority sexual orientation. Future research should investigate whether the use of APEDS is more related to the drive for muscularity and sexual orientation than symptoms of EDs.

## Figures and Tables

**Table 1 nutrients-14-04920-t001:** The reported frequency and percentage of use of protein supplements, creatine, and anabolic steroids.

	Protein Supplements	Creatine	Anabolic Steroids
At least once a week	113 (13.9%)	50 (6.2%)	1 (0.1%)
At least twice a month	33 (4.1%)	10 (1.2%)	0
At least once a month	26 (3.2%)	9 (1.1%)	0
A few times per year	41 (5.0%)	21 (2.6%)	1 (0.1%)
Less than few times/year	46 (5.7%)	43 (5.3%)	1 (0.2%)
Never	553 (68.1%)	672 (83.5%)	806 (99.6%)

Data for a few participants were not available (12 missing for protein 1.5%), 19 for creatine (2.3%), and 15 (1.8%) for anabolic steroids).

## Data Availability

Data are available upon reasonable request from the corresponding author.

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
