# Peer review of "Appearance and Performance-Enhancing Drugs and Supplements, Eating Disorders Symptoms, Drive for Muscularity, and Sexual Orientation in a Sample of Young Men"

_nutrients, 2022, doi:10.3390/nu14224920_

Round 1
Reviewer 1 Report
An interesting paper. Remarks to improve:
Title should be shortened because it is too long. Please remove “in in the general population”
The term “in the general population” and especially “men in the general population” is invalid for this study and therefore misleading. Please do not use it in any place or specify the population accurately, e.g., population of men aged x-x [but I guess cannot be supported due to lack of information whether age distribution is the same as in the population under investigation] attending xxxxx online site [the one the was used to spread invitation to participate] with a will to participate in research.
Pease specify all social media platforms that were used to spread the link to the study.
I am not sure whether it is correct to say that the prediction was stronger for the behavioral component of drive for muscularity than for its attitudinal component as the Authors do not support this notion by provide a result showing statistically significant difference between the compared coefficients [qualitative comparison may be questioned].
What is behind other sexual orientation? (23.7%)
Author Response
Dear Reviewer,
Thank you for the time and effort you have put into reading and commenting our paper.
Below, please see our responses to your comments and suggestions, point by point:
Title should be shortened because it is too long. Please remove “in in the general population”.
Response: The title has been shortened by removing ”in the general population” at the end. As we want to offer an informative title, the most important variables are still reflected in the title. To address the suggestion provided by Review #2, we added ”in a sample of young men” to the title.
The term “in the general population” and especially “men in the general population” is invalid for this study and therefore misleading. Please do not use it in any place or specify the population accurately, e.g., population of men aged x-x [but I guess cannot be supported due to lack of information whether age distribution is the same as in the population under investigation] attending xxxxx online site [the one the was used to spread invitation to participate] with a will to participate in research.
Response: Reference to the general population has been used in only two instances in the manuscript (title and abstract). In both cases, they have been deleted in the revised version.
The participants have been described in fair detail in the Methods: Participants. As this is an anonymous Internet-based study, we do agree with the reviewer that the nature of the sample in terms of generalizability remains elusive. We cannot assert that it is reflective of the general population of young males in Sweden. As described in Methods, their characteristics are similar to those of young men living in large cities in Sweden, but the point about not asserting representativity is well-taken and the text is removed.
Reviewer 2 Report
Thank you for the opportunity to review this manuscript. This is an interesting topic that can be considered by readers. Many parts of this manuscript have been very well prepared. After reviewing the manuscript, I have only some minor concerns. These are mainly, among others:
Title:
1. I wonder if it is not worth emphasizing that the study was conducted on a sample of the general male population? In my opinion, the reader can understand in the current version that this study covers the entire population.
2. Maybe it is also worth emphasizing that the study concerns a selected age group of men (e.g. young adulthood)?
Introduction:
3. In my opinion, the Introduction lacked some justification why this age group need be identified/selected in this study/analysis?
Abstract and Results:
4. I do not know if I fully understand the results, because after looking at the Abstract and the Results section, there seems to be some discrepancy (e.g. “A total of 129 participants (16.1%) reported regular use of supplements (at least once a week)” vs. “Table 1: 113 (13.9%; at least once a week)”; “while lifetime use of APEDS was reported by 32.3%.” vs. “Two hundred and fifty five participants (31.9%) reported lifetime use of protein supplement or creatine”).
Results:
5. In the records of some statistics, the appropriate markings are missing (e.g. the sign for chi square is missing; the standardized and non-standardized coefficient in regression are written with the same symbol).
6. With reference to the above comments, in regression it is difficult to verify the interpretation of the prediction power results.
7. It is not entirely clear to me why only these two predictors were selected in logistic regression (rejecting other variables, which, as shown by the authors' results, are also related to APEDS).
8. I also wonder if, if we see that APEDS may be associated with different strengths with different aspects of drive for muscularity, then if we should not analyse the subscale results in addition to the overall score in logistic regression.
Author Response
Thank you for the opportunity to review this manuscript. This is an interesting topic that can be considered by readers. Many parts of this manuscript have been very well prepared. After reviewing the manuscript, I have only some minor concerns. These are mainly, among others:
Dear Reviewer,
Thank you for the time and effort you have put into reading and commenting our paper.
Below, please see our responses to your comments and suggestions, point by point:
Title:
- I wonder if it is not worth emphasizing that the study was conducted on a sample of the general male population? In my opinion, the reader can understand in the current version that this study covers the entire population.
Response: Thank you for this comment. In response, we added ”in as sample of young men” to the title. As the Reviewer #1 pointed out, we cannot assert that the sample is fairly representative of the general population of young males in Sweden. Comparing the characteristics of your sample with the distribution of characteristics of young males from National Statistics in Sweden shows that our sample is probably very similar to those of young males living in large cities in Sweden. However, the anonymous and Internet-based nature of data make any strong assertion about the representativity of the sample impossible, although we see some fair indications of it.
- Maybe it is also worth emphasizing that the study concerns a selected age group of men (e.g. young adulthood)?
Response: To balance this comment with the wish of the Reviewer #1 who suggested a shortening of the title, we chose not to add the age span in the title. Nevertheless, we added ”in a sample of young men” to the title. In addition, the age span is provided in the abstract, as where the study was conduced (Sweden). We hope this is a suitable middle path.
Introduction:
- In my opinion, the Introduction lacked some justification why this age group need be identified/selected in this study/analysis?
Response: We appreciate this comment. In response, we have added the following to the introduction:
As the inter-quartile range for the peak age of onset for ED is 15-23 years (15), and to consider the time required to recover from ED (16) and long latency in seeking professional help for ED (17), we chose to recruit a sample of young males (15-30 years) in the current study.
Abstract and Results:
- I do not know if I fully understand the results, because after looking at the Abstract and the Results section, there seems to be some discrepancy (e.g. “A total of 129 participants (16.1%) reported regular use of supplements (at least once a week)” vs. “Table 1: 113 (13.9%; at least once a week)”; “while lifetime use of APEDS was reported by 32.3%.” vs. “Two hundred and fifty five participants (31.9%) reported lifetime use of protein supplement or creatine”).
Response: We appreciate the difficulties inherent in remembering all the details in these figures. The figure (129 &16.1%) in the abstract refer to regular use of supplements (i.e., all supplements), while 113 & 13.9%) in Table 1 is about using protein supplements. The supplements in Table 1 have been divided into proteins, Creatine and Anabolic steroids, to provide as much detail as possible.
The same applies to life-time use of APEDS in abstract (32.3%) which is per definition higher than 31.9% which is only about Proteins. They should not be the same, as APEDS in this study consists of not only protein, but also creatine and anabolic steroids. These figures are further complicated by the fact that the same individual may use several APEDS in the same time period.
Results:
- In the records of some statistics, the appropriate markings are missing (e.g. the sign for chi square is missing; the standardized and non-standardized coefficient in regression are written with the same symbol).
Response: The only instance of reporting Chi-2 in the Results is at the end of first paragraph after Table 1. Both in the word and the pdf file, the report of chi-square looks right. It has a positive value, and the abbreviation of chi-2 with Greek letter, and squared “2” looks correct. Are we missing something else?
The results of the logistic regression are now reported according to guidelines from the APA, by reporting the overall significance of the model (Chi-2 value etc.), explained portion of variance (Negelkerke R2), proportion of correctly classified cases, and odds ratio (ie, exponential B) with 95%CI, as well as beta value and p.
- With reference to the above comments, in regression it is difficult to verify the interpretation of the prediction power results.
Response: Please see our response to point #5 above. Given the magnitude of explained variance in each model, the percentage of correctly classified cases, and more clear expression of odds ratio and its confidence interval, we hope that the interpretations are now clearer.
- It is not entirely clear to me why only these two predictors were selected in logistic regression (rejecting other variables, which, as shown by the authors' results, are also related to APEDS).
Response: It is not clear to us which two predictors the reviewer is referring to. We have investigated whether APEDS can be predicted by several variables (Dietary restraint, eating concern, weight concern, shape concern, and three other sub scales of the EDE-Q based on the coding of Friborg and colleagues, binge eating, self-induced vomiting, excessive exercise, as well as drive for muscularity and its subscales). We have also investigated whether the results for dietary restraint holds across various sexual orientations. Finally we controlled the analyses for BMI and age.
- I also wonder if, if we see that APEDS may be associated with different strengths with different aspects of drive for muscularity, then if we should not analyse the subscale results in addition to the overall score in logistic regression.
Response: If we interpret this comment in a correct way, then such an analysis would be problematic due to the inherent relationship between the full DSM and its subscales. The predicting value of the subscales have been reported in the paper.